# Sleep Debt and Social Jetlag Associated with Sleepiness, Mood, and Work Performance among Workers in Japan

**DOI:** 10.3390/ijerph18062908

**Published:** 2021-03-12

**Authors:** Isa Okajima, Yoko Komada, Wakako Ito, Yuichi Inoue

**Affiliations:** 1Department of Psychological Counseling, Faculty of Humanities, Tokyo Kasei University, Tokyo 173-8602, Japan; 2Liberal Arts, Meiji Pharmaceutical University, Tokyo 204-8588, Japan; komay@my-pharm.ac.jp; 3Institute of Neuropsychiatry, Tokyo 162-0851, Japan; wakappe@gmail.com (W.I.); inoue@somnology.com (Y.I.); 4Department of Somnology, Tokyo Medical University, Tokyo 151-0053, Japan

**Keywords:** sleep debt, social jetlag, depression, work performance, sleepiness, presenteeism

## Abstract

Although sleep debt and social jetlag (SJL) influence daytime dysfunctions, the effects of both sleep debt and SJL on them have not been analyzed. The aim of this study was to examine the mutual relationship between sleep debt and SJL on daytime sleepiness, mood, and work performance. This study was a cross-sectional study on sleep health conducted on the Japanese general population. A total of 4505 general workers (30% female, aged 43.57 ± 11.63 years) were selected and analyzed. Sleep debt was defined by sleep debt index (SDI), which is the discrepancy between desired and real sleep duration. SJL and SDI scores exhibited a positive but weak coefficient (r = 0.19). In a 4 (SJL) × 3 (SDI) two-way ANOVA, the interaction effects were notable for sleepiness and depression scores, while the group effects were notable for the work performance score. For sleepiness and depression scores, SDI >2 h was not significantly different from SJL. In addition, the impact of SDI was higher than that of SJL on sleepiness (β = 0.17), depression (β = 0.16), and work performance (β = −0.10). The impact of sleep debt was more pronounced than SJL on daytime dysfunctions, although both sleep debt and SJL have negative impacts on them.

## 1. Introduction

In general, chronic short sleep duration due to sleep restriction or sleep deprivation (i.e., sleep debt) is likely to be associated with an increase in daytime sleepiness [1], an elevation of negative mood, and an impairment of job performance [2,3]. In Japan, 40–50% of adults in their 20s to 50s reported a sleep duration of ≤6 h [4].

Moreover, attention is warranted towards social jetlag (SJL), a relatively new concept that quantifies the discrepancy arising between circadian and social clocks, resulting in sleep loss and circadian misalignment [5,6]. Thirty-five percent of daytime workers in their 40s have reported an SJL of ≥1 h [7]. Thus, issues pertaining to the effects of sleep debt and SJL on daytime functions of workers need to be resolved. There has been an emphasis on the impact of sleep debt manifesting as short sleep duration on daytime impairment [3,8,9]. For example, people with <6 h of self-reported nocturnal sleep duration had a more depressive tendency than those with a sleep duration of 6–8 h, as reported in a nationwide general population survey [9]. Moreover, insufficient sleep syndrome (ISS), which is a sleep disorder characterized by an elevated daytime sleepiness due to chronic partial sleep deprivation [10], results in poorer academic performance, worse depressive mood, and more impulsivity [11].

Recent studies have shown that a larger SJL (i.e., the discrepancy of sleep phase between weekdays and weekends) contributed to the formation of behavioral problems in pre-school children [12], degradation of academic achievement and cognitive performance in adolescents [13], obesity in young adults [5,14], and depression in rural populations [15]. In addition, adolescents with ISS are likely to have a larger SJL compared to those without ISS [11]. Thus, there is sufficient evidence of the potential harmful effects of both short sleep duration and SJL on daytime functioning [3].

Incidentally, sleep duration exhibits a U-shaped association with mental and physical functions. A U-shaped association was also observed between sleep duration and depressive score [9] and between sleep duration and the risk of type II diabetes [16]. For example, Kaneita et al. [9] revealed that not only participants whose sleep duration was <6 h, but also those whose sleep duration was ≥8 h, tended to be more depressed than those whose sleep duration was 6–8 h. In addition, Matsui et al. [17] showed that sleep disturbances, physical and mental quality of life (QOL), and depression scores were worse in both the shorter and longer sleep groups compared with the group with 7–8 h of sleep. However, previous reports have not distinguished people who slept for a short/long duration due to individual life schedules from genetically determined short/long sleepers or insomniacs.

Thus, absolute sleep duration may not be sufficient to assess the sleep debt of study subjects. Considering these factors, it would be necessary to take into account the effects of both sleep debt and SJL on daytime functioning after an appropriate evaluation of sleep debt. This would also allow us to identify whether sleep debt or SJL should be the focus of improvement.

This study aimed to examine the mutual relationship between sleep debt and SJL on daytime sleepiness, mood, and work performance of the working population.

## 2. Materials and Methods

### 2.1. Participants

The study protocol was approved by the Ethics Committee of the Japanese Foundation for Sleep and Health Sciences, Tokyo, Japan. The study was conducted as part of a web-based, cross-sectional questionnaire survey on sleep health in the Japanese general population in February 2015. In this survey, participants were recruited by Rakuten Research Inc., an online marketing research company employing approximately 2.3 million Japanese individuals. An e-mail containing a link to the online questionnaire was randomly sent to individuals throughout Japan who were stratified by district, gender, and age. Participants’ age ranged from 20 to 69 years. A total of 10,000 completed responses to the questionnaire were received. Part of this pooled data has been published elsewhere [18]. Among the respondents, 4505 (3160 males, 1345 females; 43.57 ± 11.63 years) general workers who had completed the questionnaire and who had a sleep debt index (SDI) score (discrepancy between self-reported ideal and real sleep time) and SJL score of zero or more, both of which were calculated by the following method (Figure 1), were selected and analyzed. In general, the percentage with SDI <0 and SJL <0 were very low [19]. It was suggested that participants with a negative value constitute a discrete population with atypical characteristic features [20]. Based on the findings, the present study decided the exclusion criteria.

### 2.2. Assessments

#### 2.2.1. Demographic Information

The participants were asked about their age, gender, smoking habits (“Do you currently smoke?”), alcohol habits (“Do you currently have a drinking habit?”), and family constitution (“Do you currently live with your family?”).

#### 2.2.2. Japanese Version of the Epworth Sleepiness Scale (ESS)

This scale, which consists of eight items concerning the likelihood of falling asleep in sedentary situations, was used to measure subjective sleepiness. Response options for each item ranged from 0 (never) to 3 (high chance) [21].

#### 2.2.3. 12-Item Version of the Center for Epidemiological Studies Depression Scale (CES-D)

The CES-D scale measuring depressive symptoms is commonly used in epidemiological studies. The participants were asked to respond to items using a four-point Likert scale, where 0 = never or rarely, 1 = sometimes, 2 = often, and 3 = always [22].

#### 2.2.4. WHO Health and Work Performance Questionnaire-Presenteeism (WHO-HPQ Presenteeism)

This scale was used to measure absenteeism (missed days of work) and presenteeism (low performance at work which transformed to lost work-day equivalents) to generate a summarized measure of overall lost work days in a month before the interviews. The presenteeism subscale was used in this study [23,24]. Presenteeism was an 11-point Likert scale ranging from 0 (worst performance) to 10 (top performance). The final score was calculated by multiplying the raw score by 10 (0–100), in which 0 meant doing no work at all on the days spent at work and 100 meant performing at the level of a top worker.

#### 2.2.5. Sleep Habits: SDI and SJL

SDI and SJL were measured by self-reported responses to seven questions, which referred to items in the Morningness–Eveningness Questionnaire [25]. The following seven questions were used in the survey: (1) What time did you get up on a weekday, last month?; (2) What time did you get up on a weekend, last month?; (3) What time did you go to bed on a weekday, last month?; (4) What time did you go to bed on a weekend, last month?; (5) How long do you get nocturnal sleep on a weekday?; (6) How long do you sleep on a weekend?; and (7) Considering your own “feeling best of performance” rhythms, for how long would you sleep if you were entirely free for the day?

From these questions, total sleep times (TST) on both weekdays and weekends were estimated. SJL was calculated as the difference between the time of mid-sleep on weekdays and weekends [26]. In this study, we also set individual SDI, which was computed using the following formula:SDI = Self-reported ideal TST (item 7) − Real TST,
where Real TST = (TSTweekday (item 5) × 5 + TSTweekend (item 6) × 2)/7 days

### 2.3. Statistical Analysis

The data were analyzed using SPSS version 23.0 (IBM Inc., Tokyo, Japan). On the basis of the calculated time of SJL, the participants were divided into four SJL sub-categories: SJL-0 (≥0 h; range, 0 to 59 min); SJL-1 (≥1 h; range, 1 h to 1 h 59 min); SJL-2 (≥2 h; range, 2 h to 2 h 59 min); and SJL-3 (≥3 h; range, more than 3 h). The participants were also divided into three SDI sub-categories: SDI-0 (≥0 h; range, 0 to 59 min); SDI-1 (≥1 h; range, 1 h to 1 h 59 min); and SDI-2 (≥2 h; range, more than 2 h).

To compare the differences in the proportion of individuals positive for categorical variables (gender, smoking habit, drinking habit, and living alone) among SDI groups and SJL groups, the chi-square test was implemented, followed by analysis of residuals. For comparison of age among the SDI or SJL groups, one-way of analysis variance (ANOVA) was used. To take into account the impact of both sleep debt and SJL on daytime dysfunction, a 3 (SDI) × 4 (SJL) two-way ANOVA was conducted using daytime sleepiness (JESS), severity of depression (CES-D), and the degree of work performance (WHO-HPQ presenteeism) as dependent variables. When the main or interaction effects in ANOVA were observed, Bonferroni correction for p values was performed, followed by post hoc analyses. In addition, for comparison of the impact of SDI with that of SJL on the JESS, CES-D, and work performance, multiple regression analyses were conducted.

## 3. Results

### 3.1. Association of SDI and Sleep Duration with Depression, Sleepiness, and Performance

The relationship between sleep duration and depression (CES-D), daytime sleepiness (ESS), and work performance (HPQ presenteeism) was examined. When using subjective sleep duration categories, it was found to be significant for all scales (*p* < 0.001). A post hoc test showed that the ESS scores for <6 h and 6–7 h groups were lower than the scores for 7–8 h and ≥ 9 h groups, the CES-D scores for <6 h and ≥9 h groups were higher than the score for 7–8 h, and the WHO-HPQ presenteeism score for <6 h group was lower than the scores for 6–7 h and 7–8 h groups (Figure 2a,b).

As for both scales of CES-D and WHO-HPQ presenteeism, the scores were not found to be significantly different for <6 h and ≥9 h groups. On the contrary, for SDI, it was found to be significant for all scales (*p* < 0.001), and the result showed that the ≥3 h SDI group had significantly higher ESS and CES-D scores and significantly lower WHO-HPQ presenteeism score than the other SDI groups (Figure 2c,d).

### 3.2. Comparison of Demographic Data within the SDI Group and SJL Group

Table 1 shows the demographic data, means, and standard deviations for respective scales. In both SDI groups and SJL groups, significant main effects were noted for age (F_2,4502_ = 42.40, *p* < 0.01; F_3,4502_ = 71.64, *p* < 0.01). In the SDI group, the post hoc test revealed a significantly younger age in SDI-2 (≥2 h) than that in SDI-0 (≥0 h) and SDI-1 (≥1 h; both *p* < 0.01), and a significantly older age in SDI-0 than in SDI-1 (*p* < 0.01). Similarly, the post hoc test revealed that the participants in SJL-3 were significantly younger than those in the other SJL sub-categories, while those in SJL-0 (≥0 h) were significantly older than those in the other SJL sub-categories (*p* < 0.01).

Results of the chi-square test showed a significant difference in the proportion of gender in SDI sub-categories and SJL sub-categories (SDI: χ_22_ = 19.42; SJL: χ_22_ = 42.24; *p* < 0.01) and in the proportion of individuals with a smoking habit, among SJL subcategories (χ_23_ = 39.79, *p* < 0.01). As for gender, men showed a higher percentage than women for SDI-0 (28% vs. 10%, *p* < 0.01), SDI-2 (26% vs. 12%, *p* < 0.01), SJL-0 (28% vs. 10%, *p* < 0.01), and SJL-3 (≥3 h, 12% vs. 7%, *p* < 0.01). The rate of individuals with a smoking habit were lower than those without a smoking habit in SJL-0 (8% vs. 29%, *p* < 0.01), SJL-1 (≥1 h, 6% vs. 20%, *p* < 0.05), and SJL-3 (6% vs. 13%, *p* < 0.01).

### 3.3. Effect of SDI and SJL on Sleepiness

At first, we conducted a correlation analysis between SJL and SDI, and a positive coefficient was shown (r = 0.19, *p* < 0.01; Figure 3). A two-way ANOVA was performed to examine the effects of SDI and SJL on ESS. As a result, significant interaction effect (F _6,4493_ = 2.39, *p* < 0.01, η^2^ = 0.003) was noted for the scores of ESS (Figure 4a). The post hoc analysis in SJL-0 revealed a significantly higher ESS score of SDI-2 than that of SDI-0 and SDI-1 (both *p* < 0.01), and a significantly higher score of SDI-1 than that of the SDI-0 group (*p* < 0.01). In the SJL-1 subcategory, the score of SDI-2 was significantly higher than that of SDI-0 and SDI-1 (both *p* < 0.01). In SJL-2, the score of SDI-2 was significantly higher than that of SDI-0 (*p* < 0.01). Meanwhile in the SDI-0 group, the score of SJL-0 was significantly lower than that of SJL-1, SJL-2 (*p* = 0.02), and SJL-3 (*p* < 0.01, Table 2). In the SDI-1 group, the score of SJL-3 was significantly higher than that of SJL-0 (*p* < 0.01) and SJL-1 (*p* = 0.02).

The result of multiple regression analysis showed that the impact of SDI (β = 0.17, *p* < 0.001) was higher than that of SJL (β = 0.08, *p* < 0.001) on sleepiness (R^2^ = 0.04, *p* < 0.001).

### 3.4. Effect of SDI and SJL on Depression

With regard to the CES-D scores, a significant interaction effect (F _6,4493_ = 3.43, *p* < 0.01, η^2^ = 0.005) was noted (Figure 4b). The post hoc analysis revealed a significantly higher score of SDI-2 than those of SDI-0 and SDI-1 (both *p* < 0.01) and a significantly higher score of SDI-1 than that of SDI-0 (*p* < 0.01) in the SJL-0 subcategory. In both SJL-1 and SJL-2, the score of SDI-2 was significantly higher than that of SDI-0 and SDI-1 (both *p* < 0.01). In the SDI-0 group, the score of SJL-3 was significantly higher than that of SJL-0 (*p* < 0.01) and SJL-1 (*p* = 0.03; Table 2). In the SDI-1 group, the score of SJL-3 was significantly higher than those of SJL-0, SJL-1, and SJL-2 (all *p* < 0.01; Table 2).

The result of multiple regression analysis showed that the impact of SDI (β = 0.16, *p* < 0.001) was higher than that of SJL (β = 0.08, *p* < 0.001) on depression (R^2^ = 0.04, *p* < 0.001).

### 3.5. Effect of SDI and SJL on Work Performance

For the scores of WHO-HPQ presenteeism, significant main effects for the SDI group (F _2,4493_ = 3.82, *p* < 0.01, η^2^ = 0.006) and for the SJL group (F _3,4493_ = 4.25, *p* < 0.01, η^2^ = 0.003) were noted, while no significant interaction effect was found (Figure 4c). The post hoc test revealed a significantly lower score of SDI-2 than those of SDI-0 and SDI-1 for the SDI group main effect (both *p* < 0.01). For SJL group main effect, post hoc analysis also revealed a significantly lower score of SJL-3 than that of SJL-1 (*p* < 0.01). The result of multiple regression analysis showed that the impact of SDI (β = -0.10, *p* < 0.001) was higher than that of SJL (β = −0.04, n.s.) on performance (R^2^ = 0.01, *p* < 0.001).

## 4. Discussion

We conducted the present study to examine whether a U-shaped distribution of daytime function measures can be observed both in sleep duration subgroups and SDI subgroups, and to estimate the combination effect of sleep debt and SJL on daytime sleepiness, mood, and work performance.

The results showed a U-shaped distribution of depression and presenteeism among the sleep duration categories, which was similar to the results of previous studies [9,16]. On the contrary, a linear distribution was observed among the SDI groups. These findings indicate that it would be desirable to consider not only absolute sleep duration or timing, but also the individuals’ sleep needs when examining the impact of sleep debt on daytime dysfunctions. To the best of our knowledge, this is the first study to solve the aforementioned problem of U-shaped association between daytime function and sleep duration.

From the perspective of sleep debt, daytime performance was linearly deteriorated by sleep shortfalls due to personal sleep needs (i.e., SDI). In other words, daytime performance is not likely to decrease in innate short sleepers and long sleepers without any subjective insufficiency. By contrast, daytime performance of individuals with ISS, chronic insomnia, or non-restorative sleep (e.g., obstructive sleep apnea and periodic limb movements during sleep) may be deteriorated even if they sleep for a long time at night. Hence, SDI rather than sleep duration itself would be helpful in identifying levels of ones’ sleep loss.

The present study revealed that individuals with larger SJL are likely to have a worse depressive mood than those without SJL, which is consistent with a previous study revealing that misaligned circadian and social time may become a risk factor for developing depression, especially in the age range of 31–40 years [15]. The present study is the first to report that even modest circadian misalignment, which is manifested as modest SJL, may become a risk factor for inducing daytime sleepiness.

In this study, the SDI-2 group showed higher severity of daytime sleepiness, negative mood, and reduced work performance (presenteeism) as a whole. These results are quite consistent with previous findings that individuals with a short sleep duration are likely to have a higher depressive mood among the Japanese general population [9]. However, that study showed a U-shaped distribution of depressive score among respective individuals’ sleep duration categories. On the contrary, when SDI was used, the SDI score was positively correlated with the severity of depression, indicating that accumulated sleep debt may be causative for depression. Therefore, it is possible that people with long sleep duration and high depressive mood could be unsatisfied due to sleep disorders such as ISS or non-restorative sleep.

SDI × SJL interactions were shown with regard to daytime sleepiness and depression. Notably, individuals with <2 h of SDI were significantly affected by SJL in terms of the levels of daytime sleepiness and depression. However, those with SDI of >2 h had higher scores of both sleepiness and depressive mood, irrespective of the grade of SJL. In other words, sleep debt accumulation for >2 h could possibly obscure the negative effects of SJL on daytime function.

This is the first study to examine the impact of the interaction between sleep debt and SJL on workplace productivity, manifesting as presenteeism. It was found that SDI and SJL influenced the presenteeism score; specifically, workers who had sleep debt and/or sleep loss had a low productivity, indicating the influence of SJL on work productivity, which was consistent with the results of a previously reported study [13]. In addition, sleep debt had a greater impact on daytime performance than SJL, which was a manifestation of a modest circadian misalignment. These findings suggest that it would be better to address sleep debt than to address SJL. For example, when treating patients with ISS, it would be necessary to determine the discrepancy between the lengths of real and ideal sleep time where the patient can demonstrate best performance. Subsequently, the therapist must follow up with this patient to ensure minimization of SJL.

This study has several limitations. First, the subjects who participated in this study might not be representative of the Japanese general population because this study was conducted as an internet-based survey where the response rate cannot be calculated. Therefore, the causal relationship between sleep debt, SJL, and daytime dysfunction is not clear. In addition, individuals interested in the survey topic are more likely to respond to the questionnaire.

Second, the participants of this study were “workers”; however, their type of work and work schedules were not investigated. Therefore, there is a possibility that daytime workers and shift workers were both included.

Third, we operationally defined and calculated individual difference of sleep duration as a discrepancy between ideal and real sleep time for ease of convenience. To obtain a more precise difference of sleep duration, it would be desirable to use objective sleep measures such as consecutive actigraphy records for a certain period.

Finally, this cross-sectional study could not identify a causal relationship between sleep debt, SJL, sleepiness, depressive mood, and presenteeism. To clarify this, there is a need for future prospective follow-up studies to evaluate the influences of SDI and/or SJL on longitudinal change of daytime performance.

## 5. Conclusions

Both sleep debt and SJL have negative impacts on daytime sleepiness, mood, and presenteeism. However, the impact of sleep debt is more intense than that of SJL; specifically, a sleep debt of >2 h obscured the negative effects of SJL on daytime function. Therefore, it is important to improve sleep debt and SJL in order to improve work productivity. In particular, interventions on chronic sleep loss should focus on sleep debt rather than on SJL.

## Figures and Tables

**Figure 1 ijerph-18-02908-f001:**
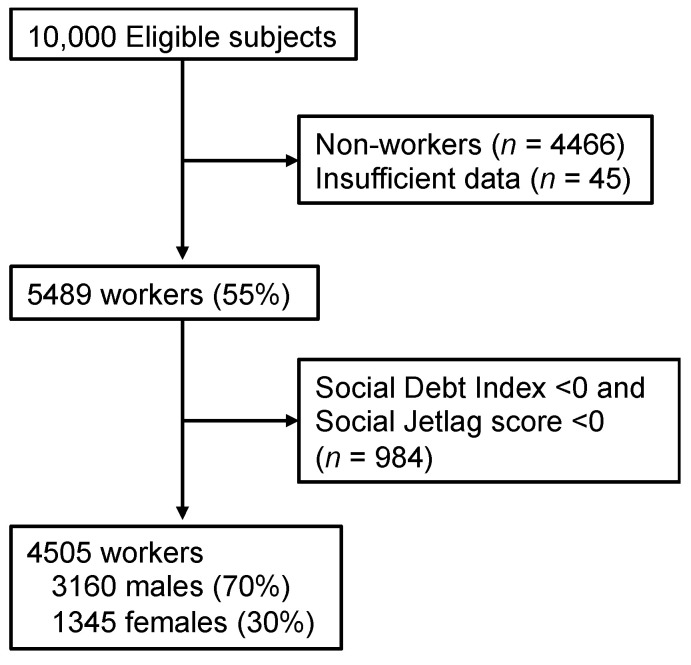
Study flowchart.

**Figure 2 ijerph-18-02908-f002:**
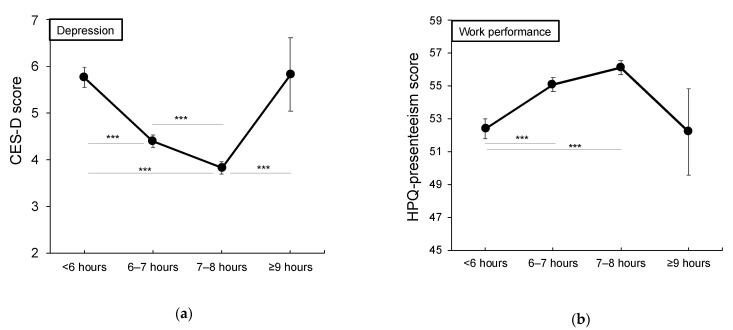
Association of sleep duration and Sleep Debt Index with depression and performance. Upper panels (**a**,**b**): Relationship between absolute sleep duration and each scale; bottom panels (**c**,**d**): Relationship between SDI and each scale. Error bars indicate the standard errors. CES-D, Center for Epidemiological Studies-Depression Scale; HPQ presenteeism, WHO Health and Work Performance Questionnaire-presenteeism subscale; SDI, sleep debt index. *** *p* < 0.001.

**Figure 3 ijerph-18-02908-f003:**
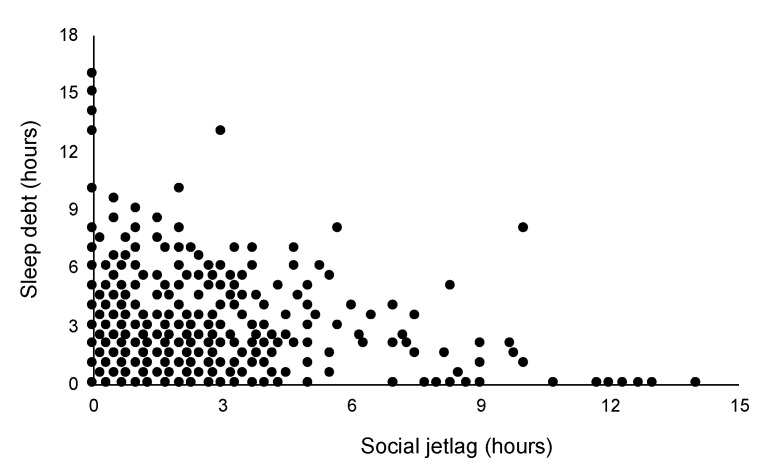
Scatter plot: Association between sleep debt and social jetlag.

**Figure 4 ijerph-18-02908-f004:**
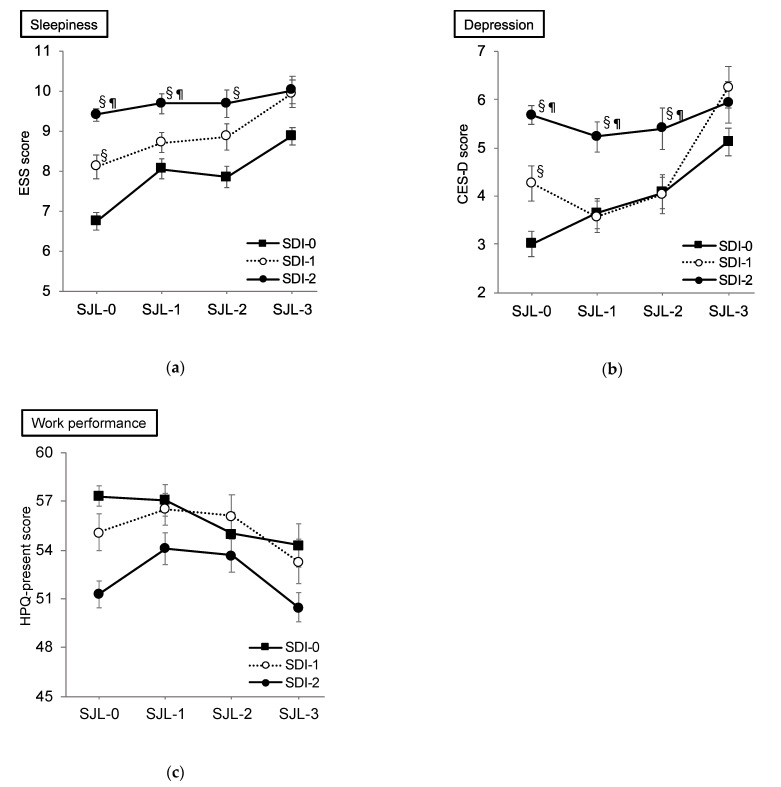
Effects of sleep debt index and social jetlag on sleepiness, depressive mood, and performance. (**a**) Effects of SDI and SJL on sleepiness; (**b**) effects of SDI and SJL on depression; (**c**) effects of SDI and SJL on work performance. CES-D, Center for Epidemiological Studies-Depression Scale; ESS, Epworth Sleepiness Scale; HPQ-presenteeism, WHO Health and Work Performance Questionnaire-presenteeism; SDI, sleep debt index; SJL, social jetlag. Error bars indicate standard errors. ^§^ In each SJL group, statistically significantly different from SDI-0 at *p* < 0.01. ^¶^ In each SJL group, statistically significantly different from SDI-1 at *p* < 0.01.

**Table 1 ijerph-18-02908-t001:** Demographic data of the sleep debt index and social jetlag sub-categories.

			Sleep Debt Index (hours)			Social Jetlag (hours)			
		Total *n* = 4505	≥0 *n* = 1693	≥1 *n* = 1087	≥2 *n* = 1725	≥0 *n* = 1702	≥1 *n* = 1157	≥2 *n* = 1157	≥3 *n* = 855
Age, (SD)		43.57 (11.63)	45.48 (12.24)	43.29 (11.40)	41.87 (10.87)	46.16 (11.98)	43.87 (11.22)	42.02 (10.80)	39.42 (10.79)
Gender, *n* (%)	M F	3160 (70) 1345 (30)	1253 (28) 440 (10)	740 (16) 347 (8)	1167 (26) 558 (12)	1273 (28) 429 (10)	816 (18) 341 (25)	538 (12) 253 (6)	533 (12) 322 (7)
Smoking habits, *n* (%)	Y N	1135 (25) 3370 (75)	411 (9) 1282 (29)	282 (6) 805 (18)	442 (10) 1283 (29)	377 (8) 1325 (29)	264 (6) 893 (20)	213 (5) 578 (13)	281 (6) 574 (13)
Drinking habits, *n* (%)	Y N	2549 (57) 1956 (43)	985 (22) 708 (16)	618 (14) 469 (10)	946 (21) 779 (17)	1001 (22) 701 (16)	641 (14) 516 (12)	432 (19) 359 (8)	281 (6) 574 (13)
Living alone, *n* (%)	Y N	871 (19) 3634 (81)	298 (7) 1395 (31)	220 (5) 867 (19)	353 (8) 1372 (31)	225 (6) 1447 (32)	215 (5) 942 (21)	166 (4) 625 (14)	235 (5) 620 (14)

M, Males; F, Females; Y, Yes; N, No.

**Table 2 ijerph-18-02908-t002:** Mean and SD of the scales for daytime function measures in each category.

	ESS		CES-D		HPQ-Presenteeism	
Groups	Mean	SD	Mean	SD	Mean	SD
SDI-0						
SJL-0	6.74	5.05	3.01	5.55	57.31	19.22
SJL-1	8.06 ^§^	4.44	3.64	5.15	57.07	19.51
SJL-2	7.85 ^§^	4.56	4.07	5.32	55.00	17.66
SJL-3	8.88 ^§^	4.87	5.13 ^§¶^	6.37	54.28	16.72
SDI-1						
SJL-0	8.11	4.49	4.27	6.61	55.09	18.99
SJL-1	8.71	4.49	3.57	5.27	56.52	19.73
SJL-2	8.87	5.03	4.04	5.62	56.11	18.35
SJL-3	9.95 ^§¶^	4.97	6.25 ^§¶†^	7.04	53.30	19.49
SDI-2						
SJL-0	9.41	5.27	5.68	7.09	51.27	20.15
SJL-1	9.70	5.03	5.23	6.32	54.10	18.63
SJL-2	9.70	4.98	5.41	6.33	53.68	18.70
SJL-3	10.33	5.03	5.94	6.52	50.46	20.81

CES-D, Center for Epidemiological Studies Depression; ESS, Epworth Sleepiness Scale; HPQ, WHO Health and Work Performance Questionnaire; SDI, sleep debt index; SJL, social jetlag. ^§^ In each SDI group, statistically significantly different from SJL-0 at *p* < 0.05. ^¶^ In each SDI group, statistically significantly different from SJL-1 at *p* < 0.05. ^†^ In each SDI group, statistically significantly different from SJL-2 at *p* < 0.05.

## Data Availability

The datasets analyzed in the current study are available from the corresponding author upon reasonable request.

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
