# Peer review of "Sleep Debt and Social Jetlag Associated with Sleepiness, Mood, and Work Performance among Workers in Japan"

_ijerph, 2021, doi:10.3390/ijerph18062908_

Round 1
Reviewer 1 Report
This paper focused on the association between social jet lag, sleep debt and individual outcomes. These are interesting issues with potentially significant consequences for health, performance, safety, and wellbeing.
Major points:
- My main concern with this paper is the overall aim – I am not sure what the value of comparing these constructs is, nor what the aim of the paper is. Based on the abstract and introduction, it is not clear what the predicted relationship between these constructs is, and why this might be of interest.
- Despite having a large sample size, once-off online survey data has significant limitations in terms of causal interpretation and bias. Furthermore, participants are described simply as being ‘workers’ – presumably there would have been significant variability in work type/schedule etc.
- Key findings of the study that are a main focus of the discussion include the impact of sleep debt on next day outcomes, and of social jetlag on next day outcomes. There has been research performed in these areas before, and I am not convinced that this study presents a novel finding. This may be clearer had the aims and research questions been outlined and defined more clearly.
Minor points:
- The results are presented in very dense paragraphs of text, suggest including some of the numbers within tables rather than in text.
- Some missing full stops and grammar issues throughout – moderate proofing required.
Author Response
Dear Reviewer #1,
This paper focused on the association between social jet lag, sleep debt and individual outcomes. These are interesting issues with potentially significant consequences for health, performance, safety, and wellbeing.
<Reply> We would like to thank you for the time you spent reviewing this manuscript. We have found your comments very useful; they have helped us to improve the manuscript significantly. Point-by-point responses have been listed below to all the comments you provided.
Major points:
- My main concern with this paper is the overall aim – I am not sure what the value of comparing these constructs is, nor what the aim of the paper is. Based on the abstract and introduction, it is not clear what the predicted relationship between these constructs is, and why this might be of interest.
<Reply> Thank you for your comments. We have revised the manuscript to address this comment.
Revised manuscript; Lines 12-15:
Although sleep debt and social jetlag (SJL) influence daytime dysfunctions, the effects of both sleep debt and SJL on them have not been analyzed. The aim of this study was to examine the mutual relationship between sleep debt and SJL on daytime sleepiness, mood, and work performance.
Revised manuscript; Lines 58-61:
Thus, absolute sleep duration may not be sufficient to assess the sleep debt of study subjects. Considering these factors, it would be necessary to take into account the effects of both sleep debt and SJL on daytime functioning after an appropriate evaluation of sleep debt. This would also allow us to identify whether sleep debt or SJL should be the focus of improvement.
- Despite having a large sample size, once-off online survey data has significant limitations in terms of causal interpretation and bias. Furthermore, participants are described simply as being ‘workers’ – presumably there would have been significant variability in work type/schedule etc.
<Reply> Thank you for your comments. We have revised the manuscript to address this comment.
Revised manuscript; Lines 266-268:
Therefore, the causal relationship between sleep debt, SJL, and daytime dysfunction is not clear. In addition, individuals interested in the survey topic are more likely to respond to the questionnaire.
Revised manuscript; Lines 269-271:
Second, the participants of this study were “workers”; however, their type of work and work schedules were not investigated. Therefore, there is a possibility that daytime workers and shift workers were both included.
- Key findings of the study that are a main focus of the discussion include the impact of sleep debt on next day outcomes, and of social jetlag on next day outcomes. There has been research performed in these areas before, and I am not convinced that this study presents a novel finding. This may be clearer had the aims and research questions been outlined and defined more clearly.
<Reply> Thank you for your comments. To clarify the novel findings of this study, we have revised the manuscript to address this comment in addition to the correction to comment 1.
Revised manuscript; Lines 62-63:
This study aimed to examine the mutual relationship between sleep debt and SJL on daytime sleepiness, mood, and work performance for the working population.
Revised manuscript; Lines 284-285:
In particular, interventions on chronic sleep loss should focus on sleep debt rather than on SJL.
Minor points:
- The results are presented in very dense paragraphs of text, suggest including some of the numbers within tables rather than in text.
<Reply> In accordance with your comment, we have deleted mean (SD) scores in the text and added them in Table 2.
- Some missing full stops and grammar issues throughout – moderate proofing required.
<Reply> The revised draft has been checked and edited again by professional editors at Editage, a division of Cactus Communications.
Reviewer 2 Report
In the present paper Okajima et al addressed the effects of sleep debt index (SDI) and Social Jet Lag (SJL) on daytime sleepiness, mood, and work performance and therelationship between SDI and SJL in producing these alteration in daily life. This issue represents a new approach to the well known consequences of sleep disorders in workers. The authors reported adequately the limitation of the study by underlying the bias on the enrolment procedure and the need of additional investigation to confirm the final observations. However, the paper needs to be improved in the results presentation, format of the tables and figures to be more readable. There are some typing errors and some missing information.
Major:
Introduction
- I suggest to add a sentence about the recent observations by Shiffer et al (IJERPH 2018) on the effects of Clockwise and Counterclockwise Job Shift Work Rotation on Sleep and Work-Life Balance on Hospital. This citation should be added at line 28, 37 and 50.
- The U-shape association described by previous studies between sleep duration and the effects on mental, physical performance and depression is an important point the needs to be more described in introduction and consequently discussed.
Materials and methods:
- Line 90 please define Likert scale and provide citation.
- I suggest inserting citations about questionnaires in the text instead in the title of sub-paragraphs 2.2.1, 2.2.2, 2.2.3
- 3 Statistics, please use hours and minutes to quantify sleep duration.
- Line 127 please explain the issues included in the 4X3 two-way ANOVA, the sentence is unclear.
Figure 2
Modify the figure legend by distinguishing “Title” from description. I suggest using “upper panels” and “bottom panel” to describe the different charts. a, b, c letters in the figure are confusing. Standard Deviation of the mean values must be shown in the charts. X axe” < 7h” should be changed in “6-7”. I suggest to use “hours” to describe X axe meaning and numbers for the sleep or sleep debt duration. To indicate the significant differences, use symbols instead letters by indicating p values for each one.
Table 1
Column titles: add (hours) i.e Sleep Debt Index (hours) and Social jetlag (hours) and delete h after the numbers.
M (SD): M should be deleted.
Increase first column and adequate the description of variables, in the present form is unclear.
The ** described in the legend is missing in the table (?)
Results.
3.2 The paragraph is complex and should be re-write to help the reader to interpret the Table. Do not use new acronyms, i.e. SJL-0, without a previous definition of this.
3.3 See above. Numbers between () and [] are confusing and make the reading difficult. It should be great to have a figure showing the correlation between SDI and SJL.
Figure 1 is inserted after figure 2, mistake? Please correct. See the same considerations and suggestions reported for Figure 2 (see above).
Discussion
Please discuss the points 1 and 2 reported in the Introduction revision (see above).
Complete data on Institutional Review Board Statement and of Informed Consent Statement
Minor:
Line 74 add “.“ after [16]
Figure 2 Legend add “.” After “performance”.
Please double check for other typing mistakes in the text, tables and figures.
Author Response
Dear Reviewer #2,
In the present paper Okajima et al addressed the effects of sleep debt index (SDI) and Social Jet Lag (SJL) on daytime sleepiness, mood, and work performance and the relationship between SDI and SJL in producing these alteration in daily life. This issue represents a new approach to the well known consequences of sleep disorders in workers. The authors reported adequately the limitation of the study by underlying the bias on the enrolment procedure and the need of additional investigation to confirm the final observations. However, the paper needs to be improved in the results presentation, format of the tables and figures to be more readable. There are some typing errors and some missing information.
<Reply> We would like to thank you for the time you spent reviewing this manuscript. We have found your comments very useful; they have helped us to improve the manuscript significantly. Point-by-point responses have been listed below to all the comments you provided.
Major:
Introduction
- I suggest to add a sentence about the recent observations by Shiffer et al (IJERPH 2018) on the effects of Clockwise and Counterclockwise Job Shift Work Rotation on Sleep and Work-Life Balance on Hospital. This citation should be added at line 28, 37 and 50. 
<Reply> Thank you for your comments. We have added the reference accordingly.
- The U-shape association described by previous studies between sleep duration and the effects on mental, physical performance and depression is an important point the needs to be more described in introduction and consequently discussed.
<Reply> Thank you for your comments. We have revised the manuscript to address this comment.
Revised manuscript; Lines 51-57:
For example, Kaneita et al. [9] revealed that not only participants whose sleep duration was < 6 h but also those whose sleep duration was ≥8 h tended to be more depressed than those whose sleep duration was 6–8 h. In addition, Matsui et al. [17] showed that sleep disturbances, physical and mental QOL, and depression scores were worse in both the shorter and longer sleep groups compared with the group with 7–8 h of sleep.
Revised manuscript; Lines 220-226:
The results showed a U-shaped distribution of depression and presenteeism among the sleep duration categories, which was similar to the results of previous studies [9,16]. On the contrary, a linear distribution was observed among the SDI groups. These findings indicate that it would be desirable to consider not only absolute sleep duration or timing but also the individuals’ sleep needs when examining the impact of sleep debt on daytime dysfunctions. To the best of our knowledge, this is the first study to solve the aforementioned problem of U-shaped association between daytime function and sleep duration.
Materials and methods:
- Line 90 please define Likert scale and provide citation.
<Reply> We have revised the manuscript to address this comment.
Revised manuscript; Lines 96-100:
The presenteeism subscale was used in this study [21,22]. Presenteeism was an 11-point Likert scale ranging from 0 (worst performance) to 10 (top performance). The final score was calculated by multiplying the raw score by 10 (0–100), in which 0 meant doing no work at all on the days spent at work and 100 meant performing at the level of a top worker.
- I suggest inserting citations about questionnaires in the text instead in the title of sub-paragraphs 2.2.1, 2.2.2, 2.2.3
<Reply> We have revised the manuscript to address this comment.
- 3 Statistics, please use hours and minutes to quantify sleep duration.
<Reply> We have revised the manuscript to address this comment.
Revised manuscript; Lines 115-119:
On the basis of the calculated time of SJL, the participants were divided into four SJL sub-categories: SJL-0 (≥0 h; range, 0 to 59 min); SJL-1 (≥1 h; range, 1 h to 1 h 59 min); SJL-2 (≥2 h; range, 2 h to 2 h 59 min); and SJL-3 (≥3 h; range, more than 3 h). The participants were also divided into three SDI sub-categories: SDI-0 (≥0 h; range, 0 to 59 min); SDI-1 (≥1 h; range, 1 h to 1 h 59 min); and SDI-2 (≥2 h; range, more than 2 h).
- Line 127 please explain the issues included in the 4X3 two-way ANOVA, the sentence is unclear.
<Reply> We have revised the manuscript to address this comment.
Revised manuscript; Lines 123-126:
To take into account the impact of both sleep debt and SJL on daytime dysfunction, a 3 (SDI) × 4 (SJL) two-way ANOVA was conducted using daytime sleepiness (JESS), severity of depression (CES-D), and the degree of work performance (WHO-HPQ presenteeism) as dependent variables.
- Figure 2
Modify the figure legend by distinguishing “Title” from description. I suggest using “upper panels” and “bottom panel” to describe the different charts. a, b, c letters in the figure are confusing. Standard Deviation of the mean values must be shown in the charts. X axe” < 7h” should be changed in “6-7”. I suggest to use “hours” to describe X axe meaning and numbers for the sleep or sleep debt duration.
<Reply> We have revised the manuscript to address these comments.
Revised manuscript; Lines 134-137:
A post hoc test showed that the ESS scores for <6 h and 6–7 h groups were lower than the scores for 7–8 h and ≥ 9 h groups, the CES-D scores for <6 h and ≥9 h groups were higher than the score for 7–8 h, and the WHO-HPQ presenteeism score for <6 h group was lower than the scores for 6–7 h and 7–8 h groups (Figure 2).
- To indicate the significant differences, use symbols instead letters by indicating p values for each one.
<Reply> We have revised Figures 2 and 4 to address this comment. In addition, the results of the interaction effects are shown in Table 2 and Figure 4, with symbols indicating the difference between the SJL groups for each SDI group in Table 2 and the difference between the SDI groups for each SJL group in Figure 4.
Table 1
- Column titles: add (hours) i.e Sleep Debt Index (hours) and Social jetlag (hours) and delete h after the numbers.
- M (SD): M should be deleted.
- The ** described in the legend is missing in the table (?)
<Reply> We have revised the manuscript to address these comments.
- Increase first column and adequate the description of variables, in the present form is unclear.
<Reply> Thank you for your comment. Since it was difficult to add more columns due to the paper size, demographic information was added to the Assessment section.
Revised manuscript; Lines 81-84:
2.2.1. Demographic information
The participants were asked about their age, gender, smoking habits (“Do you currently smoke?”), alcohol habits (“Do you currently have a drinking habit?”), and family constitution (“Do you currently live with your family?”).
Results.
- 3.2 The paragraph is complex and should be re-write to help the reader to interpret the Table. Do not use new acronyms, i.e. SJL-0, without a previous definition of this.
<Reply> Although these acronyms have been defined in 2.3 Statistical analysis (Lines 114-129), we added the manuscript to address these comments.
Revised manuscript; Lines 151-155:
In the SDI group, the post hoc test revealed a significantly younger age in SDI-2 (≥2 h) than that in SDI-0 (≥0 h) and SDI-1 (≥1 h; both p < 0.01), and a significantly older age in SDI-0 than in SDI-1 (p < 0.01). Similarly, the post hoc test revealed that the participants in SJL-3 were significantly younger than those in the other SJL sub-categories, while those in SJL-0 (≥0 h) were significantly older than those in the other SJL sub-categories (p < 0.01).
- 3.3 See above. Numbers between () and [] are confusing and make the reading difficult.
<Reply> In accordance with your comment, we have deleted mean (SD) scores from the text and added them in Table 2.
- It should be great to have a figure showing the correlation between SDI and SJL.
<Reply> In accordance with your comment, we have added a scatter plot in Figure 3.
- Figure 1 is inserted after figure 2, mistake? Please correct. See the same considerations and suggestions reported for Figure 2 (see above).
<Reply> Thank you for your comment. We have made the necessary correction.
Discussion
- Please discuss the points 1 and 2 reported in the Introduction revision (see above).
<Reply> Thank you for your comments. We have revised the manuscript to address this comment.
Revised manuscript; Lines 220-226:
The results showed a U-shaped distribution of depression and presenteeism among the sleep duration categories, which was similar to the results of previous studies [9,16]. On the contrary, a linear distribution was observed among the SDI groups. These findings indicate that it would be desirable to consider not only absolute sleep duration or timing but also the individuals’ sleep needs when examining the impact of sleep debt on daytime dysfunctions. To the best of our knowledge, this is the first study to solve the aforementioned problem of U-shaped association between daytime function and sleep duration.
- Complete data on Institutional Review Board Statement and of Informed Consent Statement.
<Reply> Thank you for your comments. We have revised the manuscript to address this comment.
Minor:
- Line 74 add “.“ after [16]
- Figure 2 Legend add “.” After “performance”.
<Reply> We have revised the manuscript to address these comments.
- Please double check for other typing mistakes in the text, tables and figures.
<Reply> The revised draft has been checked and edited again by professional editors at Editage, a division of Cactus Communications.
Round 2
Reviewer 1 Report
- Great job improving the introduction.
- P1 ln31 I’m not sure what ‘most productive years’ refers to.
- Can you please explain why individuals with SDI or SJL scores of less than 0 were removed? I would have thought it would be relevant to see how much daytime dysfunction was present for individuals without sleep problems. Also I’m not sure about the scoring of these indices, but is “< 0” the correct characterization of the respondents who were removed? Were their scores less than zero or were they zero? Or did they not complete the questionnaires?
- Minor grammar issues throughout.
Author Response
We would like to thank you for the time you spent reviewing this revised manuscript. Point-by-point responses have been listed below to all the comments you provided.
- P1 ln31 I’m not sure what ‘most productive years’ refers to.
- Can you please explain why individuals with SDI or SJL scores of less than 0 were removed? I would have thought it would be relevant to see how much daytime dysfunction was present for individuals without sleep problems. Also I’m not sure about the scoring of these indices, but is “< 0” the correct characterization of the respondents who were removed? Were their scores less than zero or were they zero? Or did they not complete the questionnaires? Minor grammar issues throughout.
<Reply>
Thank you for your comments. We have revised the manuscript to address this comment.
Revised manuscript; Lines 30-31:
In Japan, 40–50% of adults in their 20s to 50s reported a sleep duration of ≤6 h [4].
Revised manuscript; Lines 77-80:
In general, the percentage with SDI <0 and SJL <0 were very low [19]. It was suggested that participants with negative value constitute a discrete population with atypical characteristic features [20]. Based on the findings, the present study decided the exclusion criteria.
Reviewer 2 Report
I thank the Authors to have adequately addressed
all the issues requested by the reviewer. The paper have been significantly improved.
Author Response
We would like to thank you for the time you spent reviewing this revised manuscript.